UCA-YOLOv8n: a real-time and efficient fruit chunks detection algorithm for meal-assistance robot

Liu Fei felixgalaxy@163.com
Hu Mingyue
Shanghai Polytechnic University , Shanghai , China
Angiulli Giovanni
Electronic publication date: 2025 Apr 15
Publication date: 2025
Volume: 11
Electronic Location ID: e2832
Received 2024 Oct 31; Accepted 2025 Mar 24
Copyright: © 2025 Liu and Hu
Copyright year: 2025
Copyright holder: Liu and Hu
License: This is an open access article distributed under the terms of the Creative Commons Attribution License, which permits unrestricted use, distribution, reproduction and adaptation in any medium and for any purpose provided that it is properly attributed. For attribution, the original author(s), title, publication source (PeerJ Computer Science) and either DOI or URL of the article must be cited.
License URL: https://creativecommons.org/licenses/by/4.0/

Keywords: Fruit chunks detection, YOLOv8 algorithm, Universal inverted bottleneck, Coordinate attention, ADown

Funding: 2023 Shanghai Youth Science and Technology Talents Sailing Program 23YF1413800 2023 Shanghai University Young Teacher Training Grant Program ZZ202315026 This work was sponsored by the 2023 Shanghai Youth Science and Technology Talents Sailing Program (23YF1413800) and the 2023 Shanghai University Young Teacher Training Grant Program (ZZ202315026). The funders had no role in study design, data collection and analysis, decision to publish, or preparation of the manuscript.

==============================
Background

The advancement of assistive technologies for individuals with disabilities has increased the demand for efficient and accurate object detection algorithms, particularly in meal-assistance robots designed to identify and handle food items such as fruit chunks. However, existing algorithms for fruit chunk detection often suffer from prolonged inference times and insufficient accuracy.

Methods

We propose an improved YOLOv8n algorithm optimized for real-time, high-accuracy fruit chunk detection. The Universal Inverted Bottleneck (UIB) module has been integrated into the original C2f structure, significantly reducing the model’s parameter count while preserving detection accuracy. Furthermore, the coordinate attention (CA) mechanism has been incorporated into the detection head to enhance the focus on fruit chunk regions within complex backgrounds while suppressing irrelevant features, thus improving detection performance. Additionally, the ADown module from YOLOv9 has been embedded into the YOLOv8 backbone network, further increasing accuracy and reducing the number of parameters.

Results

Experimental results indicate that these enhancements substantially improve detection accuracy while reducing model size. Specifically, the optimized model achieves a 1.9 MB reduction in size, a decrease of 2.5 GFLOPs in parameter count, and an increase in mAP50 and mAP50-95 by 2.1% and 3.3%, respectively. The improved algorithm (UCA-YOLOv8n) enables real-time, accurate detection of various fruit chunks. Comparative analyses with other mainstream object detection algorithms further demonstrate the superiority and effectiveness of the proposed method.

Introduction

The rapidly evolving field of assistive technologies for individuals with disabilities has sparked significant interest in the development of intelligent meal-assistance robots (Gordon et al., 2024, 2020; Liu, Xu & Yu, 2021; Mashrur et al., 2023). These robots are designed to aid users in tasks such as identifying and handling food items. The computer vision module, equipped with advanced algorithms, accurately detects food items and facial features. Simultaneously, the control system, informed by feedback from multiple sensors, effectively guides the robotic arm to perform tasks such as selecting, retrieving, and delivering meals.

Recent advancements in food and fruit detection for feeding robots have been largely propelled by progress in artificial intelligence, machine learning, and robotics (Fan et al., 2024). A notable development is the implementation of ROS-based feeding robot systems that incorporate intention detection. For instance, Cheng, Hsiao & Chung (2020) developed a system that integrates a robotic arm with image processing technology to detect instances where a user opens their mouth continuously for a preset duration. This system subsequently feeds the user, enhancing their autonomy and dignity while alleviating the caregiver workload. Furthermore, deep learning techniques have significantly improved obstacle detection in feeding robots. Pinzón-Arenas & Jiménez-Moreno (2020) employed the Faster region-based convolutional neural network (R-CNN) neural network to identify hands as obstacles in a food assistance robot environment. Their system achieved a maximum accuracy of 77.4%, demonstrating its capability to focus on moving hands, thereby ensuring the safety and efficacy of the feeding process.

Moreover, the development of algorithms for food detection has been pivotal in enhancing feeding robot performance. Zeng et al. (2024) introduced a vision-based food handling system for meal-assistance robots, incorporating an innovative method known as Fast Image to Pose Detection (FI2PD) for food pose detection along with a closed-loop packing strategy. Their system exhibited high success rates in object recognition, pose detection, and robustness. Mohammed & Ali (2018) employed convolutional neural network (CNN) methodologies to address the challenges in food image recognition arising from the vast variety of food types and significant differences in color and texture. Chopra & Purwar (2022) utilized three distinct machine learning classification approaches (Faster R-CNN, and You Only Look Once (YOLO)) for food image recognition and classification. Their findings demonstrated that the YOLO classifier outperformed the other two methods in classification accuracy, achieving superior recognition precision. Liu et al. (2022) proposed a multi-dish food recognition method based on the EfficientDet deep learning model. Experimental results indicated that this model achieved a high mean average precision (mAP) of 0.92 in recognizing 87 different dishes. A crucial aspect of feeding systems is the accurate detection and classification of various types of food, especially fruit chunks, which present unique challenges due to their diverse shapes, sizes, and textures.

With the continuous advancement of deep learning technology, the use of deep learning algorithms for object detection has emerged as a prominent research direction (Zou et al., 2023). Studies have shown that by simulating the mechanisms of the human brain through neural networks, models can learn directly from the intrinsic features of data, demonstrating strong capabilities for image representation. Object detection algorithms such as Faster R-CNN (Ren et al., 2015), single-shot detector (SSD) (Liu et al., 2016), and YOLO (Bochkovskiy, Wang & Liao, 2020; Li et al., 2022; Redmon & Farhadi, 2017, 2018; Wang, Bochkovskiy & Liao, 2023) have been widely applied in food detection. Among single-stage object detection algorithms, the YOLO series has made significant advancements in recent years. From its initial version to the latest YOLOv8, each iteration has continuously optimized and improved detection performance. YOLOv5 is a single-stage detection model that performs object detection directly, eliminating the need for candidate boxes or region proposal networks. It employs cross-scale prediction and multi-scale fusion strategies to effectively manage diverse object scales and dense layouts. YOLOv6 introduces a novel network architecture, achieving higher detection accuracy and faster speeds through optimized network layers and parameter configurations. YOLOv7 introduces technical innovations, including network structure optimization, enhanced loss functions, and new training strategies at the algorithmic level. YOLOv8 series also demonstrates outstanding performance. Wang et al. (2023) introduced a small target connection (STC) layer into YOLOv8, which concatenates shallow and deep feature layers. They also integrated a global attention mechanism into the backbone network, leveraging feature information across dimensions to enhance detection performance. However, this approach did not fully resolve the issue of small target degradation during the feature extraction process. To address this problem, Li et al. (2023) proposed an improved YOLOv8 network with dual-channel feature fusion, incorporating the bi-pan-fpn concept to enhance the model’s small target detection capability. They also replaced several C2f modules with GhostblockV2 structures to minimize feature loss during network transmission. Despite the impressive performance of the YOLOv8 model in various detection tasks, it still faces significant challenges in detecting fruit chunks for meal-assistance robots.

In this work, we present an enhanced version of the YOLOv8n algorithm specifically designed for the real-time and efficient detection of fruit chunks in meal-assistance robots. To overcome the limitations of existing detection algorithms, we introduce several key improvements aimed at reducing inference time and increasing detection accuracy. The main contributions of this work are as follows: Firstly, the Unified Inverted Bottleneck (UIB) module from MobileNetV4 (Qin et al., 2024) is incorporated to optimize the C2f module in YOLOv8. This integration significantly reduces the parameter count while maintaining high detection accuracy, making the model more efficient for real-time applications. Secondly, the CA (Hou, Zhou & Feng, 2021) mechanism is implemented into the detection head of YOLOv8n. The CA mechanism enhances the model’s focus on relevant regions, particularly fruit chunks, within complex backgrounds. By capturing spatial and channel-wise dependencies, the CA mechanism suppresses irrelevant features and emphasizes crucial ones, thereby improving detection accuracy. Thirdly, the ADown (Wang, Yeh & Liao, 2024) module from YOLOv9 is integrated into the Conv module of the YOLOv8 backbone network. This addition further enhances detection accuracy and reduces the parameter count by efficiently downsampling features while preserving essential information.

Methods

YOLOv8n was selected for modification due to its proven balance of accuracy and computational efficiency, which are critical for real-time fruit block detection in feeding robots. Although newer versions such as YOLOv9, YOLOv10 are available, YOLOv8n remains an ideal choice for optimization due to its lightweight architecture and compatibility with embedded systems. Its end-to-end detection structure, unlike the multi-stage and computationally intensive nature of algorithms like Faster R-CNN and Mask R-CNN, allows for efficient operation with minimal hardware resources, making it highly suitable for enhancing detection capabilities in resource-constrained environments.

We present a novel fruit chunk detection method that incorporates the Unified Inverted Bottleneck (UIB) module from MobileNetV4, the coordinate attention (CA) mechanism, and the ADown module from YOLOv9 into the YOLOv8n framework. This enhanced model, termed UCA-YOLOv8n, demonstrates improved detection accuracy and a reduced parameter count compared to the original YOLOv8n.

Overview of YOLOv8 models

YOLOv8 is a state-of-the-art (SOTA) model available in various sizes—nano (n), small (s), medium (m), large (l), and extra-large (x)—designed to meet diverse application requirements. The large (l) version serves as the baseline model, with both the network depth and width set to 1. The nano (n) version reduces the network depth to 0.33 times and the width to 0.25 times that of the large version. The small (s) version also features a depth of 0.33 times but increases the width to 0.50 times the large version. The medium (m) version has a depth of 0.67 times and a width of 0.75 times that of the large version. Finally, the extra-large (x) version maintains the same depth as the large model but expands the width by 1.25 times.

This study focuses on enhancing the YOLOv8 model (Fig. 1), which comprises three primary components: the backbone for feature extraction, the neck for feature fusion, and the detection head. In both the backbone and neck, the C2f structure—designed to provide richer gradient flow—replaces the C3 structure used in YOLOv5. This improvement is achieved by adjusting the number of channels across different model scales. The Spatial Pyramid Pooling—Fast (SPPF) module efficiently extracts multi-scale features using pooling operations with varying kernel sizes, allowing the network to capture essential spatial information from multiple receptive fields without significantly increasing computational cost. In the detection head, YOLOv8 adopts a decoupled head structure, separating detection from classification tasks. Additionally, it replaces the anchor-based approach from YOLOv5 with an anchor-free method, making the model more adaptable to objects of varying shapes and sizes.

Figure 1 YOLOv8 network structure.

The proposed model architecture

This research optimizes the network architecture of the baseline YOLOv8n model. First, the original C2f structure in the backbone and neck was replaced with the C2f_UIB module, which incorporates the UIB block. This modification reduces computational complexity and the number of parameters, thereby enhancing model efficiency and performance. Second, the CA mechanism was integrated into the medium-sized target detection head of the YOLOv8n model. The CA mechanism emphasizes relevant spatial features while suppressing irrelevant background noise, improving the network’s ability to accurately detect and classify targets, as well as extract positional information. Finally, to further reduce the network’s complexity while enhancing accuracy, the downsampling component of the original backbone was replaced with the ADown module from YOLOv9. The ADown module improves target recognition and minimizes information loss, maintaining efficiency while ultimately increasing detection accuracy.

C2f_UIB

In the pursuit of enhancing real-time object detection performance, particularly in resource-constrained environments, the demand for lightweight yet powerful modules has become increasingly critical. Traditional object detection models often struggle to balance computational efficiency with detection accuracy, especially in real-world applications such as meal-assistance robots. In this study, the C2f structure of YOLOv8 is improved with the introduction of the UIB, as illustrated in Fig. 2.

Figure 2 C2f_UIB network structure.

The C2f_UIB structure integrates the UIB module into the original C2f module of YOLOv8, enhancing computational efficiency and feature extraction capabilities while maintaining a lightweight design. In the original C2f module, the input features first pass through a 1 × 1 convolution to reduce channel dimensions, minimizing computational costs while retaining critical information. The features are then split into multiple branches, each processed independently through several bottleneck layers. Each bottleneck layer consists of a 1 × 1 convolution, a 3 × 3 convolution, and an optional shortcut connection. If the shortcut is enabled, residual connections enhance gradient flow and network stability; otherwise, the layers perform pure feature transformation. The outputs of all branches are concatenated along the channel dimension and further refined through a final 1 × 1 convolution to fuse the combined features into a unified representation.

In the improved C2f_UIB structure, the bottleneck layers are replaced by UIB modules, which leverage depthwise separable convolutions and pointwise convolutions to significantly reduce parameter count and computational complexity while improving feature representation. Specifically, the depthwise convolution in the UIB module performs spatial filtering on each input channel independently, preserving spatial information while drastically reducing computational overhead. The pointwise convolution, implemented as a 1 × 1 convolution, integrates spatially filtered features across channels and adjusts the channel dimensions for optimal downstream processing. Additionally, the UIB module supports optional additional depthwise convolutions to further enhance feature extraction.

For fruit block detection, the C2f_UIB structure offers a robust solution that balances detection accuracy with computational efficiency. In this study, the C2f modules in the backbone and neck of YOLOv8n were replaced with C2f_UIB modules, as shown in Figs. 3 and 4.

Figure 3 Improved backbone structure.

Figure 4 Improved neck structure.

The C2f module ensures comprehensive feature extraction, while the UIB block reduces the model’s parameter count, making it suitable for real-time applications in meal-assistance robots. Detecting small, irregular fruit blocks in complex environments is crucial, and this structure improves detection precision while maintaining a lightweight and efficient model.

Coordinate attention

In fruit chunk detection for feeding robots, a key challenge lies in accurately identifying and localizing medium-sized objects within complex, cluttered backgrounds. While the YOLOv8n architecture is effective for real-time detection, it may struggle with medium-sized targets that are partially occluded or visually similar to other objects in the scene.

To address these challenges, the Coordinate Attention (CA) mechanism was introduced in the detection head of YOLOv8n. As illustrated in Fig. 5, the CA mechanism integrates an advanced attention block designed to enhance the model’s focus on critical spatial and channel-wise information. The input is processed via a residual connection and then split into two parallel branches: 1D horizontal global pooling and 1D vertical global pooling. These branches capture spatial information along both axes, improving the network’s perception of the object’s spatial structure. After pooling, the branches are concatenated and passed through a Conv2D layer to integrate the features. The feature map is then normalized and refined using batch normalization and non-linear activation.

Figure 5 CA mechanism structure.

Subsequently, the features are processed by two parallel Conv2D layers, followed by a Sigmoid activation function, generating attention maps that weigh the importance of features in both horizontal and vertical directions. Finally, the re-weighted feature maps are combined with the original residual input, producing a refined output that enhances the model’s attention to the spatial layout of objects.

In the context of fruit chunk detection, incorporating the CA mechanism into the last part of the medium target detection head in YOLOv8n (Fig. 6) significantly improves the model’s ability to capture both spatial and channel information, leading to enhanced detection accuracy. Unlike traditional attention mechanisms, which primarily focus on channel-wise relationships, the CA mechanism introduces spatial attention by separating global pooling operations along the horizontal and vertical dimensions. This allows the model to better recognize the spatial structure of objects, which is essential in tasks like fruit chunk detection, where object shapes and orientations vary. By leveraging CA, the model effectively highlights important features while suppressing irrelevant background noise, leading to improved localization and detection of irregularly shaped fruit chunks in complex environments. This enhancement increases the model’s ability to generalize to real-world conditions while maintaining computational efficiency, making it ideal for real-time applications in meal-assistance robots.

Figure 6 Improved head structure.

ADown module

The ADown module, as depicted in Fig. 7, improves the downsampling operation within the network by integrating multiple pooling and convolution techniques. The structure begins with an AvgPool (K = 1) operation, which smooths the input feature map. The input is then split into two parallel paths: the first path applies a Conv (K = 3, s = 2), performing both convolution and downsampling, while the second path utilizes a MaxPool (K = 2) followed by a Conv (K = 1, s = 1) to extract key features without altering the spatial dimensions. After processing through these two branches, the outputs are concatenated, producing a comprehensive feature map that captures both spatial information and critical feature details.

Figure 7 Adown network structure.

The integration of the ADown module into the YOLOv8n backbone network (Fig. 8) provides significant advantages in downsampling by combining pooling and convolution operations to retain essential spatial information. Unlike standard convolutions, the ADown module employs both average and max pooling in parallel paths, followed by distinct convolutional operations, to preserve feature richness during downsampling.

Figure 8 New backbone network structure after integrating ADown structure.

This approach enables the model to capture finer details from the input, which is crucial for tasks like fruit block detection, where objects are small and irregularly shaped. By improving the efficiency of feature extraction and retaining more spatial context, the ADown module enhances the model’s ability to distinguish fruit blocks in complex environments, resulting in better detection accuracy while maintaining computational efficiency. These improvements make the model more robust for real-time detection tasks in meal-assistance robots.

Improved YOLOv8n network structure

Figure 9 illustrates the enhanced YOLOv8n network structure, which integrates the C2f_UIB, CA, and ADown modules. The effectiveness of these components will be validated through ablation experiments in the subsequent sections.

Figure 9 Improved YOLOV8n network structure.

Experiments and evaluation metrics

Dataset

Traditional feeding robots have predominantly focused on handling easily manageable foods such as purees or soft items, catering to patients with upper limb disorders. However, these systems have largely neglected the challenge of managing more complex food items like fruit chunks, which require precise detection and handling due to their varying shapes, sizes, textures, and presentation styles.

In this study, we developed a custom dataset specifically for the task of detecting fruit chunks in meal-assistance robots. The dataset includes four distinct fruit categories: watermelon, banana, pitaya, and grape. These categories were selected to represent a diverse range of fruit types, each with unique textures, sizes, and shapes that are commonly encountered in real-world feeding scenarios. To simulate the wide variety of fruit chunk appearances and preparation methods, each fruit type was sliced into multiple shapes and sizes. The watermelon, for instance, was cut into both large and small chunks. The banana slices were prepared in different thicknesses, and the pitaya was cut into irregular shapes to reflect the diverse presentation styles found in practical applications. Grapes were selected for different sizes of samples after peeling. The dataset was meticulously annotated to include the bounding box coordinates and class labels for each fruit chunk. This annotation process was essential for training the model to accurately detect and localize fruit chunks within an image, enabling it to differentiate between various fruit types. Each fruit type in the dataset was labeled with its corresponding class: watermelon, banana, pitaya, and grape.

To enhance the diversity and robustness of the dataset, both data augmentation and sample enrichment strategies were employed. Conventional augmentation techniques, including brightness adjustment, rotation, blurring, and noise injection, were applied to simulate variations in lighting and camera perspectives. In addition, complex real-world scenarios were introduced by incorporating images with diverse backgrounds, various container types, and occlusions among fruit chunks. These enhancements aim to improve the model’s generalization and reliability in practical feeding environments. Representative samples are shown in Fig. 10.

Figure 10 Sample images from the dataset.

For training and evaluation purposes, the dataset was split into three subsets: 80% of the data was used for training, 10% for testing, and 10% for validation. This partitioning allows for comprehensive model training, performance evaluation, and validation, ensuring that the model generalizes well to unseen data. The dataset’s diversity, coupled with the extensive annotation and augmentation, ensures that the model can robustly detect and classify fruit chunks in dynamic and varied real-world feeding scenarios.

Experimental setup and parameter settings

The experiments in this study were conducted on a system running Ubuntu 22.04, using PyTorch 2.1.0 as the deep learning framework, with CUDA 12.1 for GPU acceleration. The hardware configuration included a 12 vCPU Intel(R) Xeon(R) Platinum 8336C CPU @ 2.30 GHz and two RTX 2080 Ti GPUs (22 GB total).

In our experiments, we trained the YOLOv8n model using carefully selected hyperparameters to optimize performance. The input image size was set to 640, balancing detection accuracy and computational efficiency. The model was trained over 200 epochs, providing sufficient iterations for convergence, with a batch size of 64 to allow for effective gradient updates while maintaining a manageable memory footprint. The number of data-loading workers was set to 4 to optimize the data pipeline for faster training.

For optimization, we employed the Stochastic Gradient Descent (SGD) optimizer, selected for its effectiveness in large-scale image classification tasks. The learning rate was initialized at 0.01, with a momentum of 0.937 to stabilize and accelerate convergence. Parameter groups were fine-tuned for regularization: 84 weights were set with no decay, 93 weights with a decay of 0.0005, and 93 biases with no decay. These settings were essential in balancing model complexity and generalization, resulting in a robust and efficient model.

Model comparison

The core objective of this study is to improve the performance of fruit detection in robotic feeding systems by enhancing the YOLOv8n model. To validate the effectiveness of these enhancements, a comprehensive comparison with other state-of-the-art lightweight object detection models is essential. This section provides a detailed comparison between the improved YOLOv8n and several mainstream lightweight algorithms, including YOLOv3-tiny, YOLOv5n, YOLOv5s, YOLOv7-tiny, YOLOv9-t, and YOLOv10n (Wang et al., 2024).

The comparison evaluates the models based on key metrics such as mean average precision (mAP), inference time, and model size. These metrics are critical for assessing detection accuracy, computational efficiency, and the suitability of each model for real-time applications in complex environments.

This comparative analysis aims to highlight the strengths and weaknesses of the improved YOLOv8n relative to the original YOLOv8n and other leading models. The findings will provide valuable insights into the effectiveness of the proposed enhancements and contribute to the ongoing development of more efficient and accurate object detection algorithms for robotic applications.

Evaluation metrics

The evaluation of object detection models in this study is based on a comprehensive set of metrics that offer a detailed assessment of model performance across several critical dimensions. These metrics include mean mAP, model size, parameter count, and frames per second (FPS). Each metric is crucial for understanding the trade-offs between accuracy, efficiency, and scalability, particularly in the context of deploying the models for real-time fruit detection in robotic feeding systems.

The mAP (Eq. (1)) is a standard metric in object detection that evaluates a model’s accuracy by measuring the area under the precision-recall curve. It provides a single value that summarizes the trade-off between precision and recall across different thresholds.

(1) mAP=1n∑i=1nAP(i)

where n is the number of object classes, and AP(i) is the average precision for the i-th class. Average precision (AP) is calculated by averaging the precision values across all recall levels from 0 to 1. The mAP is then obtained by averaging the AP values for all classes or across different Intersection over Union (IoU) thresholds. Precision is defined as Eq. (2), recall is defined as Eq. (3):

(2) Precision=TPTP+FP

(3) Recall=TPTP+FN

Model size refers to the storage size of the model, typically measured in megabytes (MB). It is determined by the total number of parameters and the precision with which these parameters are stored.

The number of parameters in a model refers to the total number of learnable weights and biases within the network, serving as a key indicator of the model’s complexity. It is calculated by summing the parameters across all layers.

FPS is defined as Eq. (4), it measures the speed at which a model processes images, specifically indicating how many frames (or images) the model can handle in one second. This metric is critical for real-time applications, as it directly affects the system’s responsiveness.

(4) FPS=1TimePerFrame.

Results and discussion

Ablation experiment

In this study, we enhance fruit block detection for meal-assistance robots by building on the YOLOv8n model. To achieve higher detection accuracy and real-time performance, we introduced several architectural improvements: the C2f_UIB module, the CA mechanism, and the ADown module. Each modification was systematically evaluated to determine the most effective combination, leading to the development of the YOLOv8n-C2f_UIB-CA-ADown (UCA-YOLOv8n) model, which demonstrated optimal performance.

The evaluated models are as follows, presented through a step-by-step incremental process: YOLOv8n: The baseline model.

YOLOv8n-C2f_UIB: Replace all C2f structures in YOLOv8n with C2f_UIB.

YOLOv8n-C2f_UIB-CA: Add a CA mechanism to a medium-sized object detection header.

YOLOv8n-C2f_UIB-CA-ADown (UCA-YOLOv8n): The original downsampling operations in the backbone network of YOLOv8n are replaced by the ADown modules from YOLOv9.

The loss curves (Fig. 11), representing both the training and validation phases, offer insights into the model’s convergence and overall learning efficiency. Throughout all epochs, the YOLOv8n-C2f_UIB-CA-ADown model consistently achieved the lowest box loss, indicating superior bounding box regression capabilities compared to other models. This result underscores the effectiveness of the ADown module in preserving spatial information during downsampling, which is critical for accurate localization.

Figure 11 Loss curves.

The classification loss also showed significant improvements with the introduction of each modification. The UCA-YOLOv8n model exhibited the lowest classification loss, confirming its enhanced ability to differentiate between various fruit blocks with high precision.

The distribution focal loss (DFL) curves further emphasize the advantages of the UCA-YOLOv8n model. It achieved the lowest DFL loss, indicating more accurate distribution learning for bounding box predictions, which is essential for high-quality detection in real-time applications.

The precision, recall, and mAP metrics provide a comprehensive evaluation of the detection accuracy and robustness of the models (Fig.12).

Figure 12 The results for precision, recall, and mAP.

The UCA-YOLOv8n model consistently outperformed other models in terms of precision and recall, exhibiting fewer false positives and false negatives. This is particularly important for meal-assistance robots, where reliable detection is essential for safe and efficient operation. It achieved the highest mAP at IoU = 0.5, significantly surpassing the original YOLOv8n, demonstrating its superior ability to accurately detect fruit blocks with high confidence. Additionally, the model maintained the highest mAP across a range of IoU thresholds (0.5–0.95), further confirming its robustness and consistent detection accuracy under varying conditions. This performance underscores the model’s suitability for real-world deployment, where reliability is critical.

Table 1 presents a detailed comparison of four YOLOv8n-based models, focusing on key metrics such as model size, parameter count, FLOPs, and mAP scores at two IoU thresholds (0.5 and 0.5:0.95). This analysis highlights the improvements in detection performance and computational efficiency resulting from the various model enhancements. It should be noted that CA (S, M, L) indicates the positions where the CA mechanism is applied, with S, M, and L referring to the small, medium, and large detection heads, respectively.

Table 1 Comparative analysis of ablation experiments.

Baseline
(YOLOv8n)	C2f_UIB	CA
(S,M,L)	Adown	SCDown	Model size (MB)	Parameters	FLOPs
(G)	mAP@0.5	mAP@0.5:0.95	
√					6.3	3,006,428	8.1	0.952	0.676	
√	√				4.7	2,183,836	6.1	0.954	0.683	
√	√	√(M)			4.7	2,334,644	6.3	0.964	0.7	
√	√	√(M)		√	4.3	1,907,636	5.6	0.958	0.682	
√	√	√(S)	√		4.4	2,048,658	5.6	0.967	0.691	
√	√	√(M)	√		4.4	2,051,764	5.6	0.973	0.709	
√	√	√(L)	√		4.4	2,053,916	5.6	0.964	0.688	

The original YOLOv8n is the largest and most complex model, with a size of 6.3 MB and 3,006,428 parameters. Introducing the C2f_UIB module in YOLOv8n-C2f_UIB reduces the model size by 25% to 4.7 MB and decreases the parameter count by 27% to 2,183,836, while maintaining performance. The YOLOv8n-C2f_UIB-CA model slightly increases the parameter count to 2,336,444 due to the addition of the CA mechanism, although the size remains 4.7 MB. The most optimized version, UCA-YOLOv8n, further reduces the size by 30% to 4.4 MB and parameters by 32% to 2,051,764.

In terms of computational efficiency, the original YOLOv8n requires 8.1 GFLOPs, while YOLOv8n-C2f_UIB reduces this by 25% to 6.1 GFLOPs. The YOLOv8n-C2f_UIB-CA model increases FLOPs slightly to 6.3 GFLOPs due to the CA mechanism. However, UCA-YOLOv8n is the most efficient, reducing FLOPs by 31% to 5.6 GFLOPs.

Regarding detection performance, YOLOv8n achieves an mAP of 0.952 at IoU = 0.5 and 0.676 at IoU = 0.5–0.95. YOLOv8n-C2f_UIB slightly improves this to 0.954 and 0.683, respectively. The YOLOv8n-C2f_UIB-CA model shows a more substantial boost, with mAP values of 0.964 at IoU = 0.5 and 0.7 at IoU = 0.5–0.95. Finally, the UCA-YOLOv8n model outperforms all others, achieving mAP values of 0.973 at IoU = 0.5 and 0.709 at IoU = 0.5–0.95.

To validate the effectiveness of the CA mechanism as a downsampling module, we tested the SCDown module from YOLOv10 under identical conditions in this study. As shown in rows 4 and 6 of Table 1, using the SCDown module resulted in a slight reduction in the number of parameters, but it also caused a noticeable decline in detection accuracy. Specifically, the mAP@0.5 and mAP@0.5:0.95 decreased by 1.5 percentage points and 2.7 percentage points, respectively.

In addition, to demonstrate the rationale behind integrating the CA mechanism into the medium detection head, we conducted separate tests for the small, medium, and large detection heads. As shown in the last three rows of Table 1, while the number of parameters was similar across all three configurations, the configuration with the CA mechanism in the medium detection head showed superior performance.

Overall, the UCA-YOLOv8n model not only achieves the smallest size (4.4 MB) and the lowest parameter count (2,051,764)—representing reductions of 30% and 32%, respectively, compared to the original YOLOv8n—but also demonstrates the highest computational efficiency, with FLOPs reduced by 31% to 5.6 GFLOPs. Despite these reductions in complexity and computational cost, the model delivers superior detection performance, with mAP improvements of 2.1% at IoU = 0.5 and 4.9% at IoU = 0.5:0.95. This combination of compactness, reduced computational demands, and enhanced accuracy makes the UCA-YOLOv8n model the optimal choice for real-time applications in meal-assistance robots, where both efficiency and precision are critical.

Comparison with mainstream lightweight models

In Table 2, a comparison between UCA-YOLOv8n and various mainstream lightweight models such as YOLOv3-tiny, YOLOv5n, YOLOv5s, YOLOv7-tiny, YOLOv9t, YOLOv10n, and YOLOv11n, demonstrates that UCA-YOLOv8n outperforms these models in terms of both accuracy and computational efficiency. While existing models like YOLOv3-tiny and YOLOv5s are widely used for their relatively smaller size and faster inference times, they face limitations in handling complex detection tasks with high accuracy in real-time. These models generally achieve high mAP at the cost of a significant reduction in accuracy. For example, YOLOv3-tiny achieves an mAP of only 0.968 at IoU = 0.5 and 0.699 at IoU = 0.5:0.95.

Table 2 Comparison of detection performance among mainstream lightweight algorithms.

Models	Model size (MB)	Parameters	FLOPs (G)	mAP@0.5	mAP@0.5:0.95	
YOLOv3-tiny	23.3	12,129,720	18.9	0.968	0.699	
YOLOv5n	3.9	1,764,577	4.1	0.941	0.633	
YOLOv5s	14.5	7,020,913	15.8	0.961	0.67	
YOLOv7-tiny	12.3	6,023,106	13.2	0.952	0.64	
YOLOv9t	4.7	1,971,564	7.6	0.957	0.694	
YOLOv10n	5.8	2,695,976	8.2	0.949	0.7	
YOLOv11n	5.5	2,582,932	6.3	0.972	0.708	
UCA-YOLOv8n (ours)	4.4	2,051,764	5.6	0.973	0.709	

In contrast, UCA-YOLOv8n demonstrates superior accuracy, with mAP@0.5 reaching 0.973 and mAP@0.5:0.95 improving to 0.709. These gains are attributed to the novel integration of the UIB module, CA mechanism, and ADown module within the YOLOv8n architecture. These innovations allow UCA-YOLOv8n to capture more detailed and contextual information, particularly in cluttered or occluded scenarios, while maintaining a lower computational cost. This represents a significant advancement over models like YOLOv5s (15.8 GFLOPs) and YOLOv9t (7.6 GFLOPs), where the trade-off between high accuracy and computational cost is more pronounced.

The UCA-YOLOv8n model achieves a substantial reduction in both model size (4.4 MB) and FLOPs (5.6 GFLOPs), offering a more efficient solution compared to larger models such as YOLOv3-tiny (23.3 MB, 18.9 GFLOPs) and YOLOv5s (14.5 MB, 15.8 GFLOPs). In comparison to the latest YOLOv11n, which has a model size of 5.5 MB and 6.3 GFLOPs, UCA-YOLOv8n demonstrates a superior balance between accuracy and computational efficiency. While YOLOv11n achieves comparable mAP values (0.973 at IoU = 0.5 and 0.708 at IoU = 0.5:0.95), UCA-YOLOv8n’s smaller model size and reduced FLOPs make it a more computationally efficient solution. Despite its reduced size, UCA-YOLOv8n delivers competitive performance, making it an optimal choice for real-time fruit chunk detection in meal-assistance robots, where both efficiency and accuracy are critical.

Furthermore, the innovations incorporated into UCA-YOLOv8n effectively address the limitations of earlier models in complex detection scenarios. The UIB module, which employs depthwise separable convolutions, reduces computational burden while maintaining accuracy by focusing on important spatial features. The CA mechanism enhances the model’s ability to attend to relevant areas of the image, improving detection in scenarios with complex backgrounds or occlusions. The ADown module optimizes downsampling operations, ensuring efficient feature extraction without compromising essential information.

Overall, UCA-YOLOv8n represents a significant advancement in real-time object detection, providing a balanced solution that combines high accuracy, low computational cost, and fast inference speed. This makes it a powerful tool for meal-assistance robots and other real-time detection applications.

Case analysis

Figures 13 through 18 compare the detection performance of the original YOLOv8n and the improved UCA-YOLOv8n models under various experimental conditions. These figures collectively highlight the advancements introduced by UCA-YOLOv8n in terms of detection precision, robustness, and adaptability to complex scenarios.

Figure 13 Comparison of detection results (Scene 1).

Figure 14 Comparison of detection results (Scene 2).

Figure 15 Comparison of detection results (Scene 3).

Figure 16 Comparison of detection results (Scene 4).

Figure 17 Comparison of detection results (Scene 5).

Figure 18 Comparison of detection results (Scene 6).

In Figs. 13 and 14, under standard lighting conditions, the UCA-YOLOv8n model demonstrates superior detection accuracy with higher confidence scores compared to the original YOLOv8n, underscoring its enhanced ability to recognize fruit chunks with greater precision.

Figure 15 showcases detection under scenarios with overlapping fruit chunks. UCA-YOLOv8n effectively separates overlapping objects by assigning accurate bounding boxes and avoiding misclassifications. In contrast, the original YOLOv8n shows overlapping bounding boxes and misidentifications, highlighting its limitations in resolving spatial ambiguities.

To further assess the detection performance of the proposed model, experiments were conducted in challenging scenarios involving background variations, different container types, and occlusions among fruit chunks, as shown in Figs. 16 to 18. Comparative results highlight the limitations of the original YOLOv8n model in these complex environments. Specifically, in Figs. 16 and 17, the baseline YOLOv8n model exhibits missed detections, failing to identify certain fruit chunks. In Fig. 18, in addition to missed detections, it misclassifies objects, such as incorrectly labeling a pitaya chunk as watermelon. Yellow arrows in the images indicate instances of missed detections or misclassifications. In contrast, the improved UCA-YOLOv8n model demonstrates higher detection accuracy, successfully identifying previously missed fruit chunks with greater confidence and significantly reducing false negatives and false positives. These findings emphasize the enhanced feature extraction capability, robustness, and classification precision of the proposed improvements.

Figures 19 and 20 present detection results under low-light conditions, both employing the UCA-YOLOv8n network. The left panel illustrates the model’s performance without data augmentation, while the right panel displays the outcomes of the model trained with data augmentation. This comparison clearly highlights the benefits of data augmentation in model training. The model trained with data augmentation demonstrates notable improvements in both accuracy and robustness, yielding more reliable detection results, even in challenging low-light environments.

Figure 19 Verification of the effectiveness of data augmentation methods (Scene 1).

Figure 20 Verification of the effectiveness of data augmentation methods (Scene 2).

Conclusions

This study introduces a novel enhancement to the YOLOv8n algorithm, delivering a more efficient and accurate solution for real-time fruit chunk detection in meal-assistance robots. By integrating the UIB module, CA mechanism, and ADown module, we successfully reduced the model size and parameter count, leading to a significant decrease in computational complexity. These improvements boosted detection accuracy, achieving a 97.3% mAP@0.5 and 70.9% mAP@0.5:0.95, representing a clear advancement over existing object detection algorithms such as YOLOv9t and YOLOv10n.

The optimized UCA-YOLOv8n model demonstrates both improved efficiency and performance, making it highly suitable for deployment in real-time, resource-constrained environments typical of assistive technologies. While the model excels in detecting fruit chunks, its current limitation to specific food types and the need for further testing under diverse real-world conditions highlight areas for future refinement. Expanding detection categories and enhancing adaptability and speed will be key areas for continued research. By addressing these limitations, UCA-YOLOv8n has the potential to become a more versatile tool in meal-assistance robotics and other assistive technologies, paving the way for broader applications in the field.

Supplemental Information

Supplemental Information 1 Improved engineering code for YOLOv8n.

Supplemental Information 2 Help documentation for running the code.

Additional Information and Declarations

Competing Interests

The authors declare that they have no competing interests.

Author Contributions

Fei Liu conceived and designed the experiments, performed the experiments, analyzed the data, performed the computation work, prepared figures and/or tables, authored or reviewed drafts of the article, and approved the final draft.

Mingyue Hu performed the experiments, prepared figures and/or tables, and approved the final draft.

Data Availability

The following information was supplied regarding data availability:

The project is available at figshare:Liu, Fei (2025). YOLOv8 fruit chunks detection. figshare. Journal contribution. https://doi.org/10.6084/m9.figshare.28238903.v1.

The data is available at figshare: Liu, Fei (2025). A dataset of fruit chunks. figshare. Dataset. https://doi.org/10.6084/m9.figshare.28238771.v1.

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
