# Peer review of "UCA-YOLOv8n: a real-time and efficient fruit chunks detection algorithm for meal-assistance robot"

_PeerJ Computer Science, doi:10.7717/peerj-cs.2832_

## Round 0.1 · original submission · Major Revisions

Dear Authors,
Your paper has been revised. Given the reviewers' criticism, it needs major revisions before being considered for publication in PEERJ Computer Science. More precisely, the following points must be faced in the revised version of your paper:
1) The methodology aspect needs to be thoroughly discussed, particularly the dataset and C2f_UIB procedure description;
2) more challenging samples to further valid the performance of the network must be added;
3) In Table 2, methods such as YOLOv3-tiny, YOLOv5n, and YOLOv5s are briefly compared, but there is a lack of detailed analysis of their limitations. Accordingly, the "Comparison with mainstream lightweight models" paper's section must be expanded, highlighting the advantages and innovations of UCA-YOLOv8n and explaining how these innovations address the limitations of existing methods.

Reviewer 1 ·

Basic reporting

This paper proposes an improved lightweight object detection model. By integrating the C2f_UIB, coordinate attention (CA), and ADown modules, the model achieves significant improvements in detection accuracy and parameter efficiency, demonstrating its practical value in real-time tasks. Overall, the paper demonstrates certain innovation and practical value, but major revisions are needed to enhance its academic depth, improve clarity, and expand its applicability.

Experimental design

Comparison with additional lightweight models: The paper primarily compares UCA-YOLOv8n with a few existing methods. Including more recent lightweight models (e.g., YOLOv7-tiny or other transformer-based lightweight detection methods) would better demonstrate the competitiveness of the proposed method. In addition, some state-of-the-art object detection related references should be cited and analyzed, such as: Deep learning approaches for object recognition in plant diseases: a review, Intelligence & Robotics, 2023; A Small-Object Detection Model Based on Improved YOLOv8s for UAV Image Scenarios, REMOTE SENSING, 2024.

Validity of the findings

Elaboration on ablation study: The ablation study could provide a detailed explanation of why ADown was chosen as the downsampling module, and highlight its advantages in terms of performance and efficiency compared to other downsampling modules. It would be helpful to include comparison experiments with other downsampling modules to further validate the effectiveness of the ADown module.

Additional comments

Writing and formatting improvements: In Table 2, methods such as YOLOv3-tiny, YOLOv5n, and YOLOv5s are briefly compared, but there is a lack of detailed analysis of their limitations. It is suggested to expand the "Comparison with mainstream lightweight models" section in the paper, highlighting the advantages and innovations of UCA-YOLOv8n, and explaining how these innovations address the limitations of existing methods.

Reviewer 2 ·

Basic reporting

The paper is interesting, but the provided references are limited, and the English needs improvement.

Experimental design

The methodology aspect needs to be thoroughly examined and improved, particularly the dataset and C2f_UIB description.

Validity of the findings

It is doubtful that these findings can be replicated. This is due to a lack of depth and clarity in the presentation of results and discussions. The conclusions are no different from the results and discussion section. Conclusion should provide a summary of the paper.

Additional comments

The dataset is not properly described.
The proposed algorithm is not likely to be robust for generalization.
C2f_UIB needs to be properly explained with the aid of a diagram.
The obtained results are expected having subjected the detected targets to white background.
While CA was integrated into the medium-sized target detection head, what happened to small and large?

Reviewer 3 ·

Basic reporting

The authors introduced an improved YOLOV8 network for fruit chunk detection, and experimental results showed that, based on their own dataset, the method outperformed several other mainstream lightweight models. The paper structure is good, however, the writing has to be improved:
1. citation format: for the situation at line 51, the format should be Author (Year), please correct all throughout the MS.
2. Line 47: grammar issue: .....detect when.....
3. CNN is a type of DL method, not a specific network like Faster RCNN, or YOLO. Please correct it.
4. line 78, 80, please spell out YOLO when it is introduced for the first time.
5. LIne 71 ~ 72, any data or references to support this conclusion
6 Line 89: YOLOV8 is not the latest version when the authors were writing the paper, because YOLOV9 and 10 had been released.

Experimental design

Overall, the experimental design is good. However, based on the fruit chunk samples presented in the figures, object detection for these samples is not a challenging task, because (1) there are only 4 different categories; (2) their shapes are regular, their sizes are similar; (3) no significant occlusion problem; (4) Background is simple; (5) illumination is consistent, no variation; (6) although the real size of each fruit trunk is small, the 'size' in pictures are big enough for detection because of the short working distance between the camera and the objects. I suggest adding more challenging samples to further valid the performance of the network if possible.

Validity of the findings

The results are reasonable based on the samples used in this study.

Additional comments

Please label each sub-figure and add captions for Figure 13~18.

---

## Round 0.2 · Minor Revisions

Dear Authors,
Your paper has been revised. The present version of your manuscript has been dramatically improved compared to the previous one; however, the following points need to be considered:

1) The white background of the images in the dataset raises questions about your algorithm's generalizability. You should take the appropriate tests to evaluate this point.

Reviewer 1 ·

Basic reporting

The paper is interesting, but the comparison experiments are limited.

Experimental design

The comparison experiments are limited, more state-of-the-art object detection methods should be compared and analyzed.

Validity of the findings

No further comments.

Additional comments

No further comments.

Reviewer 2 ·

Basic reporting

no comment

Experimental design

no comment

Validity of the findings

The manuscript has been greatly improved; however, the white background of the images in the dataset raises questions about the established algorithm's generalizability.

---

## Round 0.3 · Major Revisions

Dear Authors,
Your paper has been revised. It needs major revisions before being accepted for publication in PEERJ Computer Science. More precisely

1) The method's performance must evaluated on more challenging images, such as those with occlusion.

Reviewer 2 ·

Basic reporting

no comment

Experimental design

no comment

Validity of the findings

no comment

Additional comments

According to the authors, the developed model can only detect fruit chunks on a plate with a white background.

Reviewer 3 ·

Basic reporting

The study is interesting, the tools could be useful to individuals with disabilities

Experimental design

However, the method's performance should be further evaluated on more challenging images, such as those with occlusion, as suggested before. However, the authors have not addressed this comment in their response.

Validity of the findings

the findings are reasonable.

Additional comments

No

---

## Round 0.4 · Minor Revisions

Dear Authors,

Your paper has been revised. It needs minor revisions before being accepted for publication in PEERJ Computer Science. More precisely:

1) You must update the description of the dataset in the MS to reflect the newly added complex image samples.

Reviewer 3 ·

Basic reporting

No

Experimental design

I am good with the revisions. Only one minor comment: Please update the description of the dataset in the MS to reflect the newly added complex image samples.

Validity of the findings

No

Additional comments

No

---

## Round 0.5 · accepted · Accept

Dear Authors,
Your paper has been revised. It has been accepted for publication in PEERJ Computer Science. Thnk you for your fine contribution.

Reviewer 3 ·

Basic reporting

NO commnet

Experimental design

no comment

Validity of the findings

no comment

Additional comments

no comment